# Linoleate-Enrichment of Mitochondrial Cardiolipin Molecular Species Is Developmentally Regulated and a Determinant of Metabolic Phenotype

**DOI:** 10.3390/biology12010032

**Published:** 2022-12-24

**Authors:** Genevieve C. Sparagna, Raleigh L. Jonscher, Sydney R. Shuff, Elisabeth K. Phillips, Cortney E. Wilson, Kathleen C. Woulfe, Anastacia M. Garcia, Brian L. Stauffer, Kathryn C. Chatfield

**Affiliations:** 1Department of Medicine, Division of Cardiology, University of Colorado Anschutz Medical Campus, Aurora, CO 80045, USA; 2Department of Pediatrics, Division of Cardiology, University of Colorado Anschutz Medical Campus, Children’s Hospital Colorado, Aurora, CO 80045, USA; 3Division of Cardiology, Denver Health Medical Center, Denver, CO 80204, USA

**Keywords:** cardiolipin, linoleic acid, beta-oxidation, mitochondria, metabolism, heart, liver, rat, fatty acid

## Abstract

**Simple Summary:**

Cardiolipin is a very important structural phospholipid within the mitochochondrial membrane. Cardiolipin’s composition is different in various tissues but the importance of this variation has not been studied. This study aims to elucidate how mitochondrial cardiolipin composition correlates with metabolic activity in the rat heart and liver and how altering composition is sufficient to shift fatty acid use in heart cells.

**Abstract:**

Cardiolipin (CL), the major mitochondrial phospholipid, regulates the activity of many mitochondrial membrane proteins. CL composition is shifted in heart failure with decreases in linoleate and increases in oleate side chains, but whether cardiolipin composition directly regulates metabolism is unknown. This study defines cardiolipin composition in rat heart and liver at three distinct ages to determine the influence of CL composition on beta-oxidation (ß-OX). CL species, expression of ß-OX and glycolytic genes, and carnitine palmitoyltransferase (CPT) activity were characterized in heart and liver from neonatal, juvenile, and adult rats. Ventricular myocytes were cultured from neonatal, juvenile, and adult rats and cardiolipin composition and CPT activity were measured. Cardiolipin composition in neonatal rat ventricular cardiomyocytes (NRVMs) was experimentally altered and mitochondrial respiration was assessed. Linoleate-enrichment of CL was observed in rat heart, but not liver, with increasing age. ß-OX genes and CPT activity were generally higher in adult heart and glycolytic genes lower, as a function of age, in contrast to liver. Palmitate oxidation increased in NRVMs when CL was enriched with linoleate. Our results indicate (1) CL is developmentally regulated, (2) linoleate-enrichment is associated with increased ß-OX and a more oxidative mitochondrial phenotype, and (3) experimentally induced linoleate-enriched CL in ventricular myocytes promotes a shift from pyruvate metabolism to fatty acid ß-OX.

## 1. Introduction

Cardiolipin (CL) is a unique mitochondrial phospholipid that is a predominant inner-mitochondrial membrane lipid. The CL diacylglycerol backbone with four fatty acyl chains facilitates many mitochondrial enzyme functions, including those involved in respiratory chain supercomplex formation as well as oxidative phosphorylation and fatty acid beta-oxidation (ß-OX), two mitochondrial energetic processes that are inter-dependent [1,2]. Highly oxidative tissues, such as cardiac and skeletal muscle have CL that is enriched with linoleate (LA; 18:2n6) side chains whereas glycolytic tissues, like the brain, contain CL with oleate (OA; 18:1n9) and other non-LA fatty acyl side chains with little LA-enriched CL [3]. 

The heart normally derives approximately 95% of its fuel for ATP generation from oxidative sources which are primarily long-chain fatty acids, but also from pyruvate derived from glycolysis and lactate [4]. In adult human heart failure, there is a shift in cardiac fuel utilization such that there is less energy derived from oxidative sources and more from glucose and ketone metabolism [4,5,6,7]. We have shown that shifts in CL profiles are also observed in human heart failure but differ in adult and pediatric forms of disease [8]. These observations raise the question of whether metabolic substrate preference is regulated by, or co-regulated with, changes in the mitochondrial phospholipid composition. In rat models of heart failure, CL with four LA side chains (tetralineoyl CL; L_4_CL) also decreases with the progression of myocardial disease [9,10]. Additionally, loss of L_4_CL is correlated to pathological, as opposed to physiological aging, and restoration of L_4_CL with dietary intervention, restores LA-enriched CL and increases survival [11]. Another condition linking fat oxidation with LA-enriched CL is the cardio-skeletal myopathy Barth syndrome, caused by loss of the CL-remodeling enzyme tafazzin, encoded by the *TAFAZZIN* gene. In Barth syndrome, CL cannot be remodeled resulting in very low LA content on cardiac CL. We and others have shown that in both Barth patients and in mouse models of Barth Syndrome, the primary energy deficit in the heart is an inability to oxidize fats [12,13,14].

With the correlative evidence discussed above that suggest the LA content of CL modulates the oxidation of fats for fuel, we set out to investigate the direct association between CL fatty acyl content and the level of mitochondrial ß-OX. In the current study we show for the first time that the developmental shifts in CL composition are tightly correlated with levels of ß-OX in rat heart. Furthermore, dependence of ß-OX on LA-enriched CL is supported by experimental enrichment of CL with LA in neonatal rat ventricular myocytes, demonstrating the direct influence of L_4_CL on oxidative capacity in isolated cardiomyocytes.

## 2. Materials and Methods

### 2.1. Chemicals and Reagents

Unless otherwise noted, all chemicals were obtained from Sigma-Aldrich (St. Louis, MO, USA).

### 2.2. Animal Studies

All rats were in the fed state before euthanasia. Neonatal (2 days old) rats were euthanized using rapid decapitation, while juvenile (3 week) and adult (8 week) rats used for cell isolation studies were euthanized with ketamine/xylazine injection with exsanguination as a secondary method for all rats. Methods for cell isolation are detailed below. For mass spectrometry CPT, and PCR studies, neonatal rats were euthanized using rapid decapitation and juvenile and adult rats were euthanized with pentobarbital sodium (MWI Animal Health, Boise, ID, USA), heart and liver excised immediately, perfused with saline and the heart’s left ventricle (LV) was dissected (juvenile and adult; for neonatal, whole heart was used) before being flash frozen and stored at −80 °C. All animal experiments were approved by the University of Colorado Institutional Animal Care and Use Committee and were within guidelines with National Institutes of Health. 

### 2.3. Cardiolipin Quantitation

CL was quantified using previously published methods with liquid chromatography coupled to electrospray ionization mass spectrometry in an API 4000 mass spectrometer (Sciex, Framingham, MA, USA) using normal phase solvents [10]. For processing, tissue pieces were homogenized using a glass-on-glass homogenizer in PBS and lipids extracted according to previously published methods with 1 mmol tetramyristal-cardiolipin as an internal standard (Avanti Polar Lipids, Alabaster, AL, US) [10]. In cell studies, cells were scraped using PBS and lipids extracted. CL species were quantified per mg protein using a protein assay (Quick Start Bradford Protein Assay, Bio-Rad, Hercules, CA, USA) performed on tissue or cells before lipid extraction. To separate out isotope peaks, singly ionized CL species were quantified using area under the time curve for the peak ± 0.5 mass units. Individual species were then isotope corrected. For *m/z* 1424, 1426, 1450, 1452, 1454, 1456, 1474, 1476, 1498, and 1500, isotope effects from the M-2 peak were subtracted out (41.5%, 43.4%, 45.4% and 47.4% for 70, 72, 74, and 76 carbon peaks, respectively). The sum of CL species for tissue and NRVMs contained 14 different cardiolipin species: *m/z* 1422, 1424, 1426, 1448, 1450, 1452, 1454, 1456, 1472, 1474, 1476, 1496, 1498, and 1500. 

### 2.4. Real-Time quantitative PCR (RT-qPCR)

RNA was extracted using the Qiagen RNeasy mini kit (Qiagen Sciences Inc., Germantown, MD, USA) and RNA quality was verified using NanoDrop^®^ ND-1000 UV-Vis Spectrophotometer analysis (Thermo Scientific, Watham, MA, USA). SYBR Green was used to quantify gene expression using 5 ng cDNA per reaction using the StepOne RT-qPCR system (Applied Biosystems, Waltham, MA, USA). All reactions were performed in triplicate with melting curves to ensure specificity of PCR product and normalized to 18S rRNA expression. Relative gene expression was calculated using the ΔΔCT method and expressed relative to neonatal heart or liver, respectively. All primers were obtained from Integrated DNA technologies (Coralville, IA, USA) and sequences are provided in the Appendix A.

### 2.5. Carnitine Palmitoyl Transferase I and II activity

Rates of CPT I and CPT II activity in tissue homogenates and primary cardiomyocytes were quantified using a ^14^C carnitine (Perkin Elmer, Waltham, MA, USA) based radio assay [15]. The assay measures the activity of CPT I by permeabilizing the plasma membrane and measuring the production of palmitoyl carnitine from palmitoyl CoA. By permeabilizing the mitochondrial inner membrane and adding malonyl CoA to inhibit CPT I, the activity of CPT II was measured.

### 2.6. Rat Cardiac primary Cell Isolation

Neonatal ventricular myocytes (NRVMs) were isolated from ventricles of 2-day-old Sprague Dawley rats (Charles River Laboratories, Wilmington, MA, USA) by enzymatic digestions as described previously [16]. Briefly, cells were isolated by trypsin digestion from the ventricles of neonatal rats and cells were plated at a concentration of 1.5 × 105 cells per well in 12-well tissue culture gelatin-coated plates in medium containing 5% bovine calf serum. After 24 h, the medium was changed to serum-free medium containing insulin, transferrin, bovine serum albumin, vitamin B12, and penicillin; all medium solutions were buffered with HEPES (pH 7.5) to a final concentration of 20 mM. 

Juvenile rat left ventricular myocytes (JRVMs) were isolated from 3-week-old sex-matched Sprague Dawley rats (74–83 g). This protocol was modified from previously published methods [17,18]. The heart was removed, cannulated by the aorta, and retrograde perfused with perfusion buffer for 4 minutes at 37 °C (120.4 mM NaCl, 14.7 mM KCl, 0.6 mM KH_2_PO_4_, 0.6 mM Na_2_HPO_4_, 1.2 mM MgSO_4_, 4.6 mM NaHCO_3_, 30 mM Taurine, 10 mM 2,3-butanedione monoxime (BDM), 5.5 mM Glucose, pH 7.2). After perfusion, the heart was digested with 1.4 mg/mL Collagenase II (Roche) for 11 to 14 min followed by mincing in Collagenase II containing 8 mM Ca^2+^ and pipetting up and down for 5 min. Digestion was stopped by adding perfusion buffer with fetal bovine serum and the slurry filtered through a sterile 100µm mesh. Myocytes were separated from non-myocytes by centrifugation at 400xg for 4 min at room temperature. The myocytes were resuspended in perfusion buffer and layered over 6.45 mg/mL of bovine serum albumin (BSA) and allowed to settle for 15 min. Cells were resuspended in DMEM (Gibco) supplemented with BSA (2 mg/mL), 2 mM carnitine, 5 mM creatine, 5 mM Taurine, BDM (1 mg/mL), and penicillin-streptomycin-L-glutamine (100 µg/mL) at a concentration of 90,000 cells/mL and plated in 6-well laminin-coated plates at a density of 225 to 250 cells/mm2. After, 48 h, cells were scraped, collected in PBS, and stored at −80 °C.

Adult rat left ventricular myocytes (ARVMs) were isolated in accordance with previously published methods [18]. Briefly, ARVMs were isolated from >8-week-old, female Sprague Dawley rats (250–300 g). Hearts were retrograde perfused with perfusion buffer as above and digested with 2 mg/ml Collagenase II for 20–24 min. The right ventricle was removed and the left ventricle minced in 2 mg/ml Collagenase II with 8 mM Ca^2+^ and the slurry filtered using 100 µm mesh. Cells were then handled as JRVMs above. The ARVM culture was maintained in DMEM supplemented with BSA (2 mg/ml), carnitine (2 mmol/L), creatine (5 mmol/L), taurine (5 mmol/L), BDM (1 mg/mL), and penicillin-streptomycin (100 μg/mL) and plated at concentration of 60,000 cells/mL and plated in 6-well laminin-coated plates at a density of 150 to 200 cells/mm2. After 48 h, cells were scraped, collected in PBS, and stored at −80 °C until used for experimentation.

### 2.7. Alteration of CL Fatty Acyl Side Chains in NRVMs

NRVMs used for Seahorse and some CL experiments were treated with either (final concentration) 10 µM BSA alone or 10 µM BSA with either 50 µM dioleoylphosphidylglycerol (O_2_PG) or 50 µM dilineolylphosphatidylglycerol (L_2_PG) for 48 h according to the methods of Corrado et al. [19]. Both species of phosphotidylglycerol (PG) were from Avanti Polar Lipids (Alabaster, AL, USA). Cells were washed and either scraped off with PBS for CL assays or a Seahorse assay was performed.

### 2.8. Agilent Seahorse

The Agilent Seahorse XF Fatty Acid Oxidation (FAO) Protocol was adapted in our laboratory for use on NRVMs. Briefly, NRVMs were seeded in a 96-well XFe Seahorse cell plate at a density of 50,000 cells per well, in replicates of 3–5 wells per condition. NRVMs were treated for 48 h with either BSA, L_2_PG or O_2_PG as described above. The day prior to the assay, a calibration plate (Agilent Seahorse XFe96 Sensor Cartridge (Agilent Technologies, Santa Clara, CA, USA) was hydrated with sterile nano-H_2_O at 37 °C in a non-CO_2_ incubator overnight. Additionally, cells were treated 24 h before the assay with (0.5 mM) carnitine to assure that FA transport into mitochondria was not rate limited. The day of the assay, the H_2_O in the calibration plate was changed to Seahorse calibrant solution (Agilent Technologies, Santa Clara, CA, USA). One hour prior to the assay, the cells were gently rinsed twice with phosphate-buffered saline (PBS, ThermoFisher Scientific, Waltham, MA, USA), after which warmed Seahorse FAO assay media was added (DMEM plus 0.5 mM L-Carnitine and 5 mM HEPES; pH 7.4) and cells incubated for at least 45 min at 37 °C in a non-CO_2_ incubator to de-gas. Etomoxir (60 µM) or vehicle (H_2_O) was added to appropriate wells 30 min before the assay. The assay medium was also used for the preparation of the inhibitors. Freshly diluted inhibitors were pre-loaded in the calibration plate in three separate ports for a final well concentration of: 3 µM oligomycin, 2 µM carbonyl cyanide-4-(trifluoromethoxy)phenylhydrazone (FCCP), and 1.8 µM each of rotenone/antimycin A. Immediately before the assay, cells were treated with saturating concentrations of either BSA (to promote carbohydrate based oxidation) or Palmitate -conjugated to BSA (to promote fatty acid based oxidation) (Agilent Technologies, Santa Clara, CA, USA). All values were normalized to relative live cell number using the CyQUANT™ Direct Cell Proliferation Assay (ThermoFisher Scientific, Waltham, MA, USA). Mitochondrial oxygen consumption rate (OCR) was calculated as pmol/min/cells (Agilent Seahorse Wave Software 2.0.6) after normalization to the average OCR of each individual prep day.

### 2.9. Statistical Analysis

Data analysis used Prism version 8 (GraphPad Software, La Jolla, CA, USA). Effects were analyzed with *p* < 0.05 being significant and *p* < 0.1 reported as trending toward significance. Data were tested for Gaussian distribution with D’Agostino & Pearson omnibus and the Shapiro–Wilk normality test. For non-normally distributed data sets, data was transformed by log2. Statistical analysis for experiments using group analysis, such as RT-qPCR and cell treatment experiments, were done using a One ANOVA test with *p* < 0.05 being significant and *p* < 0.1 reported as trending toward significance. Graphs with bars show standard error of the mean. For normally distributed data sets, Tukey’s post hoc analysis was performed to assess group differences; for non-normally distributed data, the Šídák’s multiple comparisons test was used.

## 3. Results

### 3.1. Cardiolipin Is Altered Proportionally with Age in Heart but Not Liver

The CL mass spectrometry spectra showing various groups with side chain lengths varying from 68 carbons (C) to 78 C are shown in Figure 1. CL shifts with increasing age in the heart (Figure 1A–C), but, in contrast, the spectra remains relatively stable with age in liver (Figure 1D,F). The largest shift in adult rat heart compared to neonatal or juvenile rats is the predominance of the 72 C side chain CL peak, largely comprised of four 18-carbon fatty-acyl chains. Within this peak, CL is primarily composed of different combinations of LA and OA. Liver does not demonstrate this dramatic change in the 72 C peak with age. When quantified in heart, the absolute quantity (Figure 1G), percentage of CL composed of L_4_CL (Figure 1H), the L_4_CL/tetraoleoylcardiolipin (O_4_CL) ratio (Figure 1I), and the sum of CL species (Figure 1J) all increase significantly with age, whereas in liver, if these amounts differ, they tend to peak in juvenile rats (Figure 1K–N). 

### 3.2. Genes Encoding Fatty Acid Metabolism and Glycolysis Show Changes with Age and Organ Type That Correlate with Cl Alterations

To determine if these age-related changes in CL spectra correlate with metabolic changes in the developing heart and liver, gene expression profiles were determined for genes that encode proteins associated with fatty acid vs. glucose metabolism (Figure 2). 

Genes associated with fatty acid metabolism include those coding Acyl-CoA Medium and Long Chain Dehydrogenases (*MCAD*, *ACADL* Figure 2A–D), Fatty Acid Translocase (*CD36*, Figure 2E,F), and the Fatty Acid Transporter Protein 1 (*SLC27A1*, Figure 2G,H). *MCAD*, *ACADL*, and *SLC27A1* all significantly increase with age in heart with *CD36* trending toward an increase. *ACADL* and *CD36* significantly decrease with age in liver and the others are relatively unchanged. In contrast, the Glucose Transporter Type 1 (*SLC2A1*, Figure 2I,J) shows a decrease in the heart with increasing age, but no change in the liver. Additional genes assessed are shown in Appendix A and primer sequences for these genes show in Appendix A. including genes related o cardiolipin biosynthesis and remodeling; *TAFAZZIN* expression is unchanged in both heart and liver with age, while cardiolipin synthase (*CRLS1*) increases in the adult liver, but is unchanged in the heart. Trifunctional protein subunit A (*HADHA)* mRNA expression decreases with age in heart and liver, whereas subunit B (*HADHB*) has increased expression in heart but declines in liver. Other mitochondrial genes such as carnitine acylcarnitine translocase (*SLC25A20*) and pyruvate dehydrogenase E1alpha subunit (*PDHA1*) are unchanged with age. Primer sequences for these genes are shown in Appendix A. 

### 3.3. Carnitine Palmitoyltransferase Activity and Expression in the Heart and Liver

Both tissue CPT I and CPT II enzyme activities increase with age in the rat heart and decrease with age in liver (Figure 3A–D). The gene *CPT1a*, a neonatally expressed gene, decreases in both heart and liver with age, while *CPT1b* decreases only in liver and *CPT2* expression is constant across age (Appendix A).

### 3.4. Cardiolipin and CPT Patterns Are Recapitulated in Primary Cultures of Neonatal, Juvenile, and Adult Cardiac Myocytes

To validate the use of primary cells instead of myocardial tissue, primary rat ventricular myocytes from neonatal heart (NRVM), juvenile heart (JRVM) or adult heart (ARVM) were isolated and cultured, and CL analysis was performed. The CL spectra for these three cell types are shown in Figure 4A–C and the ratio of L_4_CL to O_4_CL is shown in Figure 4D. CPT activity was quantified in Figure 4E,F, where CPT I activity did not significantly increase with age. (Figure 4E), but CPT II activity significantly increased (Figure 4F). The spectra and ratios have a similar pattern to those of whole tissue in Figure 1, although in cultured cells, there tends to be more oleate in CL derived from the media, reflected in a decrease in the L_4_CL/O_4_CL ratio from over 400 in the tissue to 55 in cells (Figure 1I vs. Figure 4D).

### 3.5. Cardiolipin Side Chains Are Experimentally Altered in NRVMs to Vary LA and OA Content

Dietary supplementation in rats can influence cardiac CL composition, with LA-enriched diets resulting in L_4_CL-enriched cardiac mitochondria [11,20]. Likewise, treatment of NRVMs with either dioleoylphosphatidylglycerol (O_2_PG, with two oleate side chains) or dilineoylphosphatidylglycerol (L_2_PG with two lineolate side chains), resulted in a shift of the 72 C CL spectra either to the right (OA enrichment) or left (LA enrichment), respectively [19]. Figure 5A–C shows the 72 carbon CL spectra of NRVMs treated with BSA (Figure 5A), OA enriched NRVMs (Figure 5B) or LA enriched NRVMs (Figure 5C). Figure 5D shows the overlay of both OA enriched and LA enriched spectra which results in a shift to the right or left, respectively. The percent of each of the 72 carbon CL species is quantified in Figure 5E where the dominant species in the OA enriched cells is L_2_O_2_CL and in the LA enriched cells is L_4_CL. The ratio of L_4_CL/O_4_CL is shown in Figure 5G.

### 3.6. Changing the CL Profile Alters Oxygen Consumption Using Palmitate as a Substrate

Experimentally altering the CL side chain composition in NRVMs, either to an OA-enriched CL profile or an LA-enriched CL profile resulted in changes to mitochondrial respiration as measured by the Seahorse bioanalyzer. The degree of these differences depended on the fuel source: either pyruvate or palmitate. Oxygen consumption rate (OCR) for ATP production with all three treatments: BSA, OA enrichment or LA enrichment using either pyruvate or palmitate as a substrate are shown in Figure 6A. The CPT inhibitor etomoxir is used to prevent entry of palmitate into the mitochondria and therefore prevent oxidation of palmitate. In both pyruvate and palmitate-driven respiration, myocytes with CL that are LA-enriched have the highest OCR, and addition of etomoxir eliminates the influence of LA-enriched CL to support respiration. Maximal respiration (shown in Figure 6B) is also influenced by LA enrichment, which supports higher levels of oxidative metabolism compared to OA-enriched CL and BSA controls, and this effect is eliminated with etomoxir treatment. Using pyruvate as a substrate, the LA enriched cells maintain a basal level of fatty acid metabolism that is inhibited with etomoxir (Figure 6B). A Seahorse trace of one representative plate of treated cells is shown in Figure 6C without etomoxir and in Figure 6D in the LA enriched cells with palmitate as a substrate plus or minus etomoxir.

## 4. Discussion

The human heart is known to undergo a gradual bioenergetic maturation during prenatal and early postnatal development where it becomes increasingly dependent on fats for energy [21], but this study is the first to tie that bioenergetic shift to a transition in CL composition from the neonatal to juvenile and then to a mature adult phenotype in the rodent heart. In contrast to the heart, liver in the fed state derives most of its energy through glycolysis and pyruvate oxidation [22]. Although CL has been previously linked to mitochondrial function in general [23,24], this is the first demonstration of a dependence on the LA-enrichment of CL to support maximal oxidative metabolism through fatty acid ß-OX in cardiac myocytes. Mitochondrial CL profiles are developmentally regulated in the heart demonstrating the importance of ß-OX to support a consistent pool of cardiac ATP. By contrast, the liver, which is also highly metabolic, does not depend on ß-OX in its fed state and does not have developmentally altered CL composition. Gene expression analysis in these two tissues also support this premise that the neonatal heart shifts to a profile favoring ß-OX in adulthood, while in liver ß-OX enzymes largely decrease or remain unchanged with maturity into adulthood. 

Metabolic enzyme expression in heart was correlated with enzyme activities of fatty acid transport proteins CPT I and CPT II in the heart and liver with increasing age. The increase in enzyme activities were seen in the heart exclusively, suggesting a greater reliance on fatty acids as an energy source in the adult myocardium (Figure 3). Analysis of gene expression shows profound transcriptional changes with divergent energetic profiles. In heart, the ß-OX transcripts *MCAD*, *ACADL*, *CD36* (coding for Fatty Acid Translocase), and *SLC27A1* (coding for Fatty Acid Transporter 1) increased with age, whereas in liver, they were unchanged with age. *CD36*, a gene that encodes fatty acid translocase and facilitates cellular uptake of fatty acids [25] decreased in liver. The pattern of these gene expression changes suggests age-dependent up-regulation of constituents of the fatty acid ß-OX pathway in heart, with either no change or a down-regulation with age in liver. In heart, there was reduced expression with age of *SLC2A1*, the gene that encodes a subunit of the *GLUT1* transporter, implying a reduced need to uptake glucose into the cardiomyocyte [26]. In contrast to the metabolic genes’ expression changes discussed above, there is no apparent developmental regulation of genes that are critical for CL synthesis (cardiolipin synthase, *CRLS1*) or remodeling (*TAFAZZIN* and *HADHA*; Appendix A), yet there are clearly developmental changes in the CL profile (Figure 1). This observation suggests that CL synthesis and remodeling genes have constitutive expression in heart and liver with age, necessary for ongoing mitochondrial biogenesis and turnover. Developmental changes in CL profiles must be due to other mechanisms which could include post-translational enzyme regulation, enzyme compartmentalization, phospholipase activities, or other mechanisms. This is an area of interest for future study.

In order to justify the use of primary cultured cardiomyocytes to further studies related to CL remodeling and ß-OX, it was determined that cardiomyocytes isolated from neonatal, juvenile, and adult rats recapitulate the aforementioned CL shifts (Figure 4). Though CPT I activity in the cardiomyocytes appeared relatively unaffected with age (Figure 4E), CPT II activity was higher in ARVMs in comparison to NRVMs and JRVMs further supporting the conclusion that these primary cultured cells maintain the metabolic phenotype observed in whole tissue correlates (Figure 4F). 

In order to demonstrate a causal relationship between LA enrichment of CL side chain composition and mitochondrial reliance on ß-OX and oxidative metabolism, CL in NRVMs was experimentally altered such that CL profiles were either LA- or OA-enriched. This alteration was accomplished by incubation of PG containing either two LA or two OA side chains. PG is a direct precursor of CL, which is synthesized when PG and CDP-diacylglycerol form CL by the enzyme CL synthase. This method of altering CL diminishes the likelihood of non-CL effects that could result from adding fatty acids directly to alter CL. The dominant species of CL with BSA treatment alone in neonatal cells is L_3_O or L_2_O_2_ with L_4_CL comprising just 20.5% of the 72 C species. With L_2_PG treatment of cardiomyocytes, LA is incorporated into CL and the predominant CL species become L_3_O and L_4_CL (L_4_CL: 33.8%; Figure 5E). With O_2_PG treatment, the predominant species become L_2_O_2_ and LO_3_.with L_4_CL only comprising 13.8% of species. These relatively small shifts in LA enrichment in the fatty acyl profile of CL produce dramatic effects on both coupled and uncoupled mitochondrial respiration, especially using palmitate as a substrate (Figure 6A,B). This increase in oxygen consumption rate in the LA enriched cells is negated if the CPT inhibitor etomoxir is added, confirming that this altered respiratory activity is driven by and dependent on mitochondrial fatty acid ß-OX. It is also interesting to note that the sum of all CL species does not seem to have an effect on ß-OX in NRVMs. Even with less total CL, LA-enriched CL has much higher rates of palmitate induced oxidation than BSA treated or OA-enriched cells (Figure 5G and Figure 6A,B).

This work sheds new light on how alterations in the CL molecular species profiles may be necessary and/or sufficient to shift mitochondrial metabolism toward fatty acid oxidation and augmented oxidative metabolism to supply ATP for fuel in the heart specifically. Evidence to support this dependence of ß-Ox on CL molecular species profile can be demonstrated in heart failure where there is a decrease in LA containing CL and an increase in the dependence on glycolysis for energy [5,9]. This is also demonstrated in Barth Syndrome where the almost complete absence of LA on CL leads to a dramatic decrease in fatty acid oxidation [12,13,14]. Physiologic differences in tissues also support this hypothesis, with an example being brain, an organ that relies on glycolysis for the majority of its energy needs and has almost no detectable L_4_CL [3].

These observations suggesting that LA-enrichment promotes and regulates fatty acid utilization may provide a potential therapeutic opportunity by supporting ß-OX and oxidative metabolism in cardiac diseases via modulation of the CL profile with diet or pharmacologic agents that promote CL remodeling. Indeed, our prior work has shown that the proportion of L_4_CL could readily be increased through LA dietary supplementation in Spontaneously Hypertensive Heart Failure rats, and in human subjects ingesting LA-enriched cookies, presenting a potential therapeutic strategy to an energy-starved failing heart [11,20,27].

## 5. Conclusions

In summary, this work shows the importance of CL in the tissue-specific regulation of the metabolic profile of mitochondria. The significance of CL side-chain composition is evident in its developmental regulation and the ability of alterations in side-chain composition to modulate ß-Ox and mitochondrial respiration. These unique attributes point to a critical role of this phospholipid and its regulation on mitochondrial function. This characterization is important for understanding metabolic phenotypes in human development and disease and may reveal new therapeutic targets in pathologic states including genetic cardiomyopathies and acquired forms of myocardial disease, such as ischemic heart failure. Yet, more is to be learned about other roles of CL, and the relevance of composition, chain length, and symmetry [28] on parameters such as tissue specificity, mitochondrial dynamics, and intracellular signaling.

## Figures and Tables

**Figure 1 biology-12-00032-f001:**
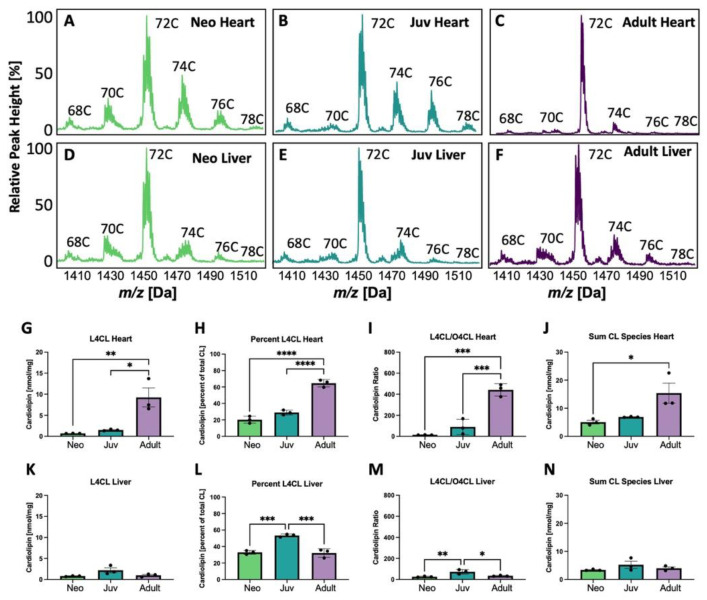
Cardiolipin (CL) changes with age in heart and liver. (**A**–**F**) Cardiolipin (CL) spectra in heart and liver from neonatal (Neo), juvenile (Juv), and adult rat heart and liver. C indicates the total number of side chain carbons on the CL in (**A**–**F**). Tetralineoyl CL (L_4_CL) levels (**G**,**K**) and percent (**H**,**L**) are quantified in heart and liver, respectively. Quantification of the ratio of L_4_CL to tetraoleoyl CL (O_4_CL) in heart (**I**) and liver (**M**) and the sum of CL species in heart (**J**) and liver (**N**). In (**G**–**N**), * *p* < 0.05, ** *p* < 0.01, *** *p* < 0.001, **** *p* < 0.0001; *N* = 3 separate animals; error bars represent standard error of the mean.

**Figure 2 biology-12-00032-f002:**
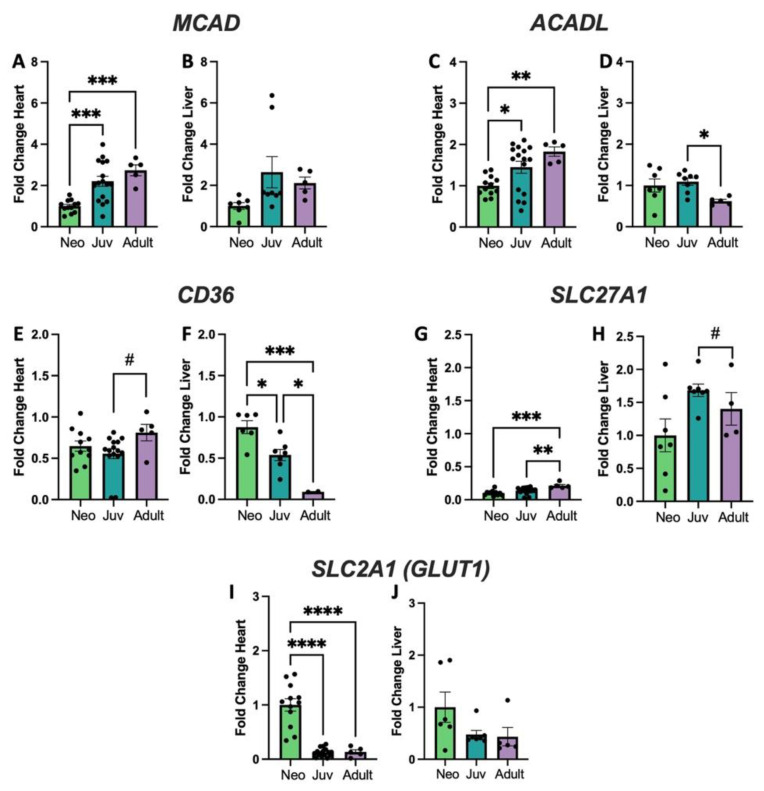
Genes with differential expression in heart and liver. Quantitative rt-qPCR for the genes *MCAD*, coding for Medium Chain Acyl CoA Dehydrogenase (**A**,**B**); *ACADL*, coding for Long Chain Acyl CoA Dehydrogenase (**C**,**D**); *CD36* coding for Fatty Acid Translocase (**E**,**F**); *SLC27A1*, coding for Fatty Acid Transport Protein 1 (**G**,**H**), and *SLC2A1* coding for the Glucose Transporter Type 1(*GLUT1*) (**I**,**J**). # *p* < 0.01, * *p* < 0.05, ** *p* < 0.01, *** *p* < 0.001, **** *p* < 0.0001; *N* = 5–16 separate animals. Error bars represent standard error of the mean.

**Figure 3 biology-12-00032-f003:**
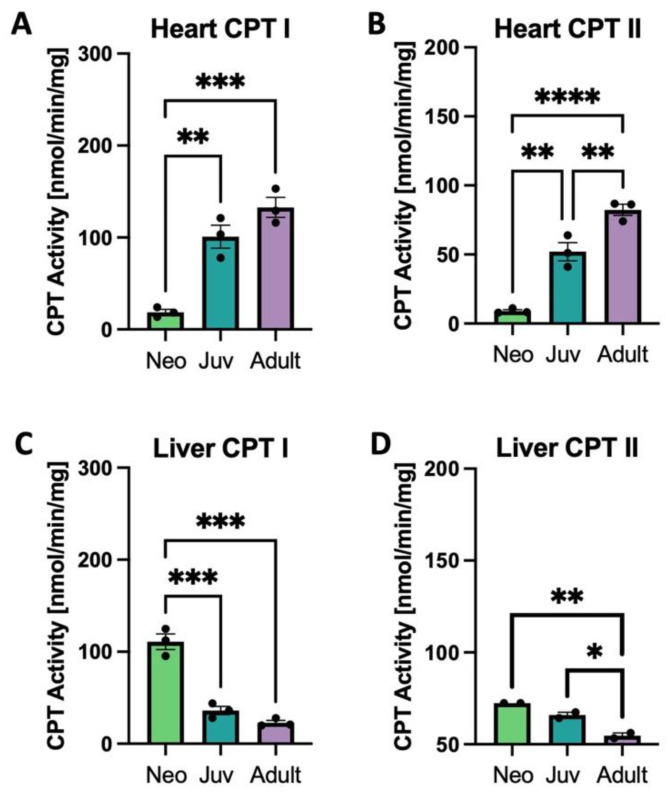
Carnitinepalmitoyltransferase (CPT) activity in heart and liver. CPT I and II activity in rat heart (**A**,**B**) and liver (**C**,**D**); * *p* < 0.05, ** *p* < 0.01, *** *p* < 0.001, **** *p* < 0.0001; *N* = 3 separate animals. Error bars represent standard error of the mean.

**Figure 4 biology-12-00032-f004:**
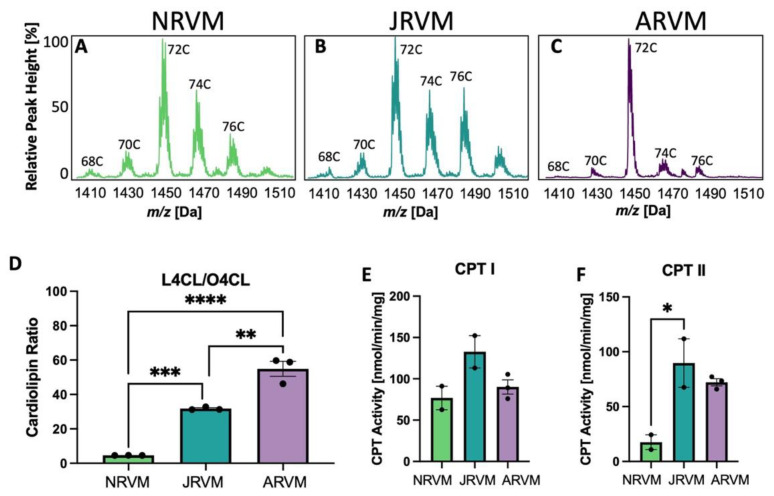
CL and CPT changes in primary cultured rat heart cells. (**A**–**C**) CL spectra in heart and liver from neonatal (**A**), juvenile (**B**), or adult (**C**) rat ventricular myocytes (NRVM, JRVM, ARVM, respectively). C indicates the total number of side chain carbons on the CL in (**A**–**C**). (**D**)The ratio of L_4_CL to O_4_CL in these cells; (**E**,**F**) CPT I and II activity in these cells. * *p* < 0.05, ** *p* < 0.01, *** *p* < 0.001, **** *p* < 0.0001. In (**D**–**F**), *N* = 2-3; error bars represent standard error of the mean.

**Figure 5 biology-12-00032-f005:**
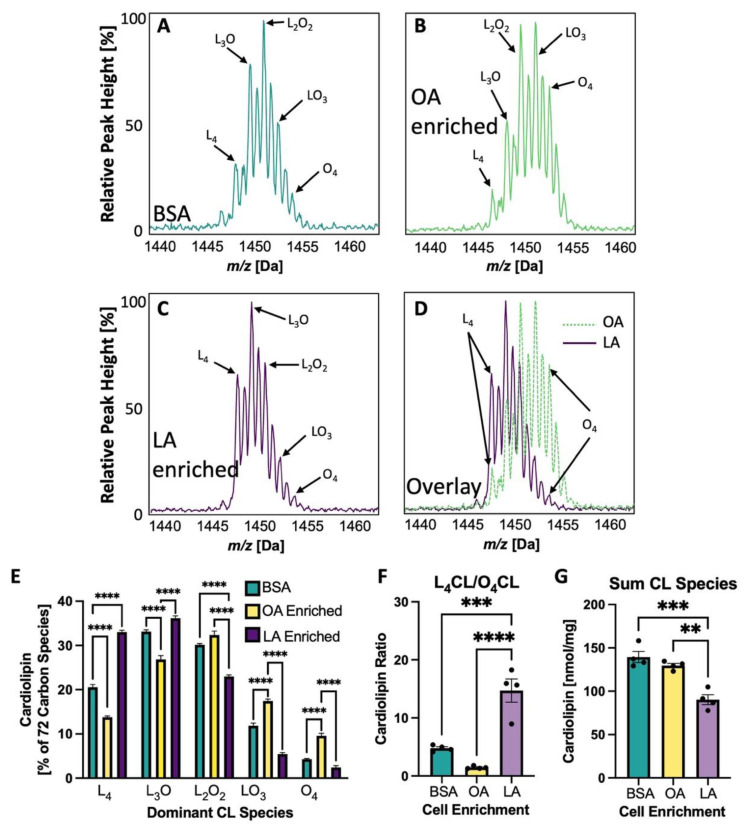
CL Spectral Shift with LA or OA Enrichment. (**A**–**D**) CL spectra showing only 72 carbon peaks having five predominant CL species with variable numbers of linoleate(L;LA) or oleate(O;OA) side chains. (**A**) Neonatal rat ventricular myocytes (NRVMs) treated with bovine serum albumin (BSA) alone, (**B**) NRVM CL enriched with OA, (**C**) NRVM CL enriched with LA, (**D**) overlay of (**B**,**C**). (**E**) The percentage of each of the five 72 carbon CL species with these treatments. (**F**) The L_4_CL to O_4_CL ratio for each treatment. (**G**) The sum of 14 detectable CL species. ** *p* < 0.01, *** *p* < 0.001; **** *p* < 0.0001, *N* = 4 in (**E**) through (**G**). Error bars represent standard error of the mean.

**Figure 6 biology-12-00032-f006:**
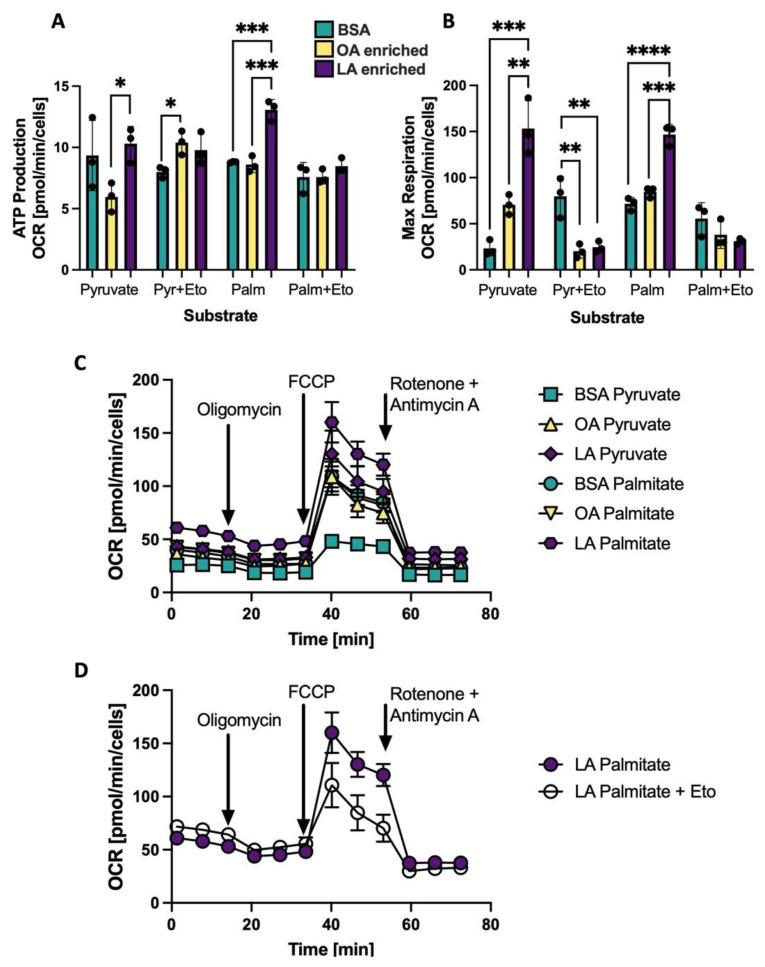
Mitochondrial oxygen consumption rate (OCR) is shifted depending on CL composition and substrate composition. (**A**) ATP Production OCR using either pyruvate (Pyr) or palmitate (Palm) as a substrate with and without the CPT inhibitor etomoxir (Eto) in NRVMs treated with BSA, OA enriched or LA enriched for 48h to alter CL composition. (**B**) Maximal respiration same conditions as (**A**). (**C**) Representative Seahorse spectra for the samples without Eto. (**D**) Representative Seahorse spectra in LA enriched cells using palmitate as a substrate with and without Eto. * *p* < 0.05, ** *p* < 0.01, *** *p* < 0.001, **** *p* < 0.0001; *N* = 3 separate Seahorse plates from separate NRVM preparations. Error bars represent standard error of the mean.

## Data Availability

The data presented in this study are available on request from the corresponding author.

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
