# Peer review of "Linoleate-Enrichment of Mitochondrial Cardiolipin Molecular Species Is Developmentally Regulated and a Determinant of Metabolic Phenotype"

_biology, 2022, doi:10.3390/biology12010032_

Round 1

Reviewer 1 Report

Sparagna et al. describe the developmental regulation of cardiolipin content and profile in heart tissue and primary cells isolated from myocardial tissue of neonatal, juvenile and adult rats. The authors show that in heart but not liver there is an increase in L4Cl and L4Cl/O4Cl ratio and this is associated with increased CPT I activity. Moreover, the authors efficiently modulate cardiolipin composition by culturing the cells in presence of different forms of PG with an increase in basal and maximal respiration when cells are cultured in presence of LA enriched medium.

The manuscript is sound in experimental setting and design and the conclusions are supported by the results. Therefore, this reviewer suggests the publication after minor changes and correction are applied to figure 3, which is partially cut in the current version of the manuscript.

Author Response

Reviewer #1:

The manuscript is sound in experimental setting and design and the conclusions are supported by the results. Therefore, this reviewer suggests the publication after minor changes and correction are applied to figure 3, which is partially cut in the current version of the manuscript.

We appreciate the reviewer’s supportive comments and have corrected Figure 3.

Reviewer 2 Report

The manuscript by Sparagna and Jonschner et. al. provides evidence for a correlation between development-associated changes in cardiolipin (CL) fatty acyl content and changes in metabolic substrate utilization in cardiac tissue. It is known that metabolic substrate utilization within the heart shifts from glycolysis/pyruvate oxidation to fatty acid/beta-oxidation during early postnatal development. The potential link to CL fatty acyl content is novel and this reviewer warmly welcomes the manuscript and finds it well executed. Nonetheless, some issues should be addressed:

1)      No details relating the methods for quantification of CL species are included in the manuscript. Details should be included related to the mass spectrometry methodologies used. Also, how was the overlapping isotope distributions of the CLs species accounted for? (i.e. for bar graphs depicted in Figures 1, 4, 5)

2)      Is the lack of a change in CRLS1 and TAFAZZIN gene expression surprising to the authors? Perhaps this warrants discussion in the text?

3)      It is tempting to speculate that the total fatty acyl content in the heart is also changing during development, did the authors consider quantifying total fatty acyl content? Or total fatty acyl content of isolated mitochondria?

4)      The left side of Figure 3 is cut off.

5)      Figure 5 legend is missing a description of panel G.

6)      In Figure 6C it is unclear which Seahorse tracings are with and without etomoxir.

Author Response

Reviewer #2 requests the following issues to be addressed: 

  • No details relating the methods for quantification of CL species are included in the manuscript. Details should be included related to the mass spectrometry methodologies used. Also, how was the overlapping isotope distributions of the CLs species accounted for? (i.e. for bar graphs depicted in Figures 1, 4, 5)

We thank the reviewer for a very insightful observation.  Cardiolipin is an unusually large phospholipid and requires subtraction of isotopes in order to identify specific species. Although this was discussed years ago in the original analysis paper, we agree that it is useful to review the method of analysis here and have added the following details on the isotope subtraction in the methods section: “To separate out isotope peaks, singly ionized CL species were quantified using area under the time curve for the peak ± 0.5 mass units.  Individual species were then isotope corrected.  For m/z 1424, 1426, 1450, 1452, 1454, 1456, 1474, 1476, 1498, and 1500, isotope effects from the M-2 peak were subtracted out (41.5%, 43.4%, 45.4% and 47.4% for 70, 72, 74, and 76 carbon peaks respectively).  The sum of CL species for tissue and NRVMs contained 14 different cardiolipin species: m/z 1422, 1424, 1426, 1448, 1450, 1452, 1454, 1456, 1472, 1474, 1476, 1496, 1498, and 1500.”

  • Is the lack of a change in CRLS1 and TAFAZZIN gene expression surprising to the authors? Perhaps this warrants discussion in the text?

We appreciate the reviewer’s comment about the lack of change in TAFFAZIN and CRLS1 gene expression in the heart, which warrants further discussion in the manuscript.  We have commented on these results in section 3.2, and have added comment in the discussion section that provides an alternative hypothesis for how cardiolipin remodeling may occur in a developmentally-regulated manner (second paragraph of discussion, page 15).

It is tempting to speculate that the total fatty acyl content in the heart is also changing during development, did the authors consider quantifying total fatty acyl content? Or total fatty acyl content of isolated mitochondria?

This is an excellent suggestion, and we are pursuing this line of research, not in bulk measurement of fatty acids (such as done with gas chromatography mass spectrometry), but by measuring fatty acyl-CoAs, reflective of the content of intra-mitochondrial fatty acids, or acyl-carnitines (the form necessary for transport across the mitochondrial membrane).  These lipid analyses methods are in use in our lab but require a higher sensitivity mass spectrometer or large amounts of sample material (for acyl CoA quantification in particular). Inclusion of this experiment and analysis was beyond the scope of this current study and is certainly an area of interest for further work. There was not an obvious section within the discussion to mention this, but can certainly include mention of this topic at the reviewers request.

  • The left side of Figure 3 is cut off.

Thank you for noting this formatting error, which has been corrected.

  • Figure 5 legend is missing a description of panel G.

We thank the reviewer for noting this omission. We have fixed the legend to include the description of 5G.

  • In Figure 6C it is unclear which Seahorse tracings are with and without etomoxir.

We thank the reviewer for this observation.  We originally did not include the conditions with etomoxir in them in panel 6C because there would be too many tracings in one figure. Based on this reviewer comment we decided to include an additional panel containing just the condition treated with palmitate as a substrate with and without etomoxir.
